# Fast and Near-Optimal Algorithms for Private Hypothesis Selection

**Hilal Asi** [1]  **Hongjie Chen** [2]

## Abstract

We study the problem of private hypothesis selection: given samples from an unknown distribution drawn from a finite hypothesis class, the goal is to identify the best hypothesis under the constraint of differential privacy. Existing algorithms for this problem are either computationally expensive or achieve sub-optimal statistical rates. We propose new algorithms that achieve near-optimal rates while running in nearly linear time in the number of hypotheses. Rather than applying the exponential mechanism directly with a score function that requires pairwise comparisons between hypotheses, our approach introduces a carefully designed loss function based on a small set of strong hypotheses. This structure allows the score to be evaluated efficiently for most hypotheses, yielding significant computational savings. We further extend our algorithms to the agnostic setting, where the true distribution may not belong to the hypothesis class. As an application, we obtain faster differentially private algorithms for universal statistical estimation in low dimensional settings.

## 1. Introduction

Hypothesis selection is a fundamental problem in statistics and learning theory. Given $n$ samples $x_1, \ldots, x_n$ drawn from an unknown distribution $H$ and a finite collection of candidate distributions (hypotheses) $\mathcal{H} = \{H_1, \ldots, H_m\}$, the goal is to identify the hypothesis that best explains the data. This problem underlies many classical tasks such as density estimation, model selection, and goodness-of-fit testing, and has been extensively studied (Yatracos, 1985; Devroye & Lugosi, 1996; 1997; Mahalanabis & Stefankovic, 2008; Aliakbarpour et al., 2024).

In modern applications, however, datasets are sensitive and

therefore ensuring that the algorithm does not leak private information is critical. This has motivated the study of private hypothesis selection, where the learner must identify a good hypothesis under the constraint of differential privacy. Incorporating privacy constraints fundamentally changes the problem, requiring algorithms to carefully balance statistical accuracy, privacy guarantees, and computational efficiency.

Several approaches to private hypothesis selection have been proposed in the literature (Canonne et al., 2018; Bun et al., 2021; Aden-Ali et al., 2021; McMillan et al., 2022; Asi et al., 2024; Aliakbarpour et al., 2025). While (Canonne et al., 2018; McMillan et al., 2022) designed a simple, efficient, and optimal clamped log-likelihood algorithm for the binary case $m = 2$, the problem is more challenging when $m$ is large. Prior work (Bun et al., 2021; Aden-Ali et al., 2021; Asi et al., 2024) has studied the non-binary setting and proposed new algorithms. However, these algorithms are based on Scheffé sets and tournament-style comparisons that require comparing every pair of distributions, resulting in computationally prohibitive $O(m^2 \cdot n)$ runtime.

More recently, (Aliakbarpour et al., 2025) developed nearly linear-time algorithms for private hypothesis selection. While these algorithms significantly improve computational efficiency, their statistical guarantees are not optimal. In particular, for simple hypothesis selection where $H \in \mathcal{H}$, their algorithms require $n \geq \Omega(\log^3(m)/(\Delta_{\mathrm{TV}}^2 \varepsilon))$ samples, in contrast to the standard private Scheffé sets algorithm (Bun et al., 2021) which only requires $n \geq \Omega(\log(m)/(\Delta_{\mathrm{TV}} \varepsilon))$ where $\Delta_{\mathrm{TV}}$ is the smallest total variation distance between two distributions in $\mathcal{H}$.

These limitations lead to our central question: is it possible to design private hypothesis selection algorithms that simultaneously achieve near-optimal statistical rates and nearly linear computational complexity? Despite significant progress, this question has remained open in several important settings.

### 1.1. Our results

In this paper, we make progress on the above question and obtain the following results:

**Near-linear and near-optimal algorithms.** For the simple hypothesis selection problem where $H \in \mathcal{H}$, we pro-

---

[1]Apple [2]ETH Zurich. Part of the work is done while the author was an intern at Apple.

*Proceedings of the 43$^{rd}$ International Conference on Machine Learning*, Seoul, South Korea. PMLR 306, 2026. Copyright 2026 by the author(s).

*Table 1.* Summary of results for simple hypothesis selection under $\varepsilon$-DP. We let $\Delta_{\mathrm{TV}}(\mathcal{H}) := \min_{P,Q \in \mathcal{H}: P \neq Q} \|P - Q\|_{\mathrm{TV}}$ and $\Delta_\varepsilon$ is defined in (2).

| Result | Sample complexity $(n)$ | Time complexity |
| --- | :---: | :---: |
| Private Scheffé tournament (Bun et al., 2021) | $O\big(\frac{\log m}{\Delta_{\mathrm{TV}}^2} + \frac{\log m}{\Delta_{\mathrm{TV}}\varepsilon}\big)$ | $O(m^2 \cdot n)$ |
| Private $T$-mechanism (Asi et al., 2024) | $O\big(\frac{\log m}{\Delta_\varepsilon}\big)$ | $O(m^2 \cdot n)$ |
| (Aliakbarpour et al., 2025) | $\tilde{O}\big(\frac{\log^3 m}{\Delta_{\mathrm{TV}}^2 \varepsilon}\big)$ | $\tilde{O}(m \cdot n/\Delta_{\mathrm{TV}})$ |
| This work (Theorem 4.1) | $O\big(\frac{\log m}{\Delta_\varepsilon}\big)$ | $O\big(m \cdot n\big)$ |

pose a new algorithm that runs in time $O(m \cdot n)$ and obtains sample complexity $O\left(\frac{\log m}{\Delta_\varepsilon}\right)$. This matches the sample complexity of the private $T$-mechanism (Asi et al., 2024). Moreover, for the binary case $m = 2$, our algorithm recovers the optimal rate $1/\Delta_\varepsilon$ as in (Canonne et al., 2018).

Our approach builds on the tournament-based framework underlying the private Scheffé tournament algorithm (Bun et al., 2021), replacing the total variation distance with an appropriate divergence as in (Asi et al., 2024). Unlike prior methods that require exhaustive pairwise comparisons, our key contribution is a two-stage algorithmic design that significantly reduces computational cost.

In the first stage, we efficiently and privately identify a small set of strong hypotheses capable of eliminating a large fraction of the hypothesis class. To do so, we sample a small random subset of $\mathcal{H}$ and use the AboveThreshold mechanism to privately identify a hypothesis that outperforms all hypotheses in the subsample. The size of the subset ensures that any such hypothesis must eliminate all but a small fraction of hypotheses in $\mathcal{H}$.

In the second stage, we construct a carefully designed loss function that (i) can be evaluated efficiently for most hypotheses, (ii) has bounded sensitivity, and (iii) closely approximates the original pairwise loss for sufficiently strong hypotheses. Applying the exponential mechanism to this loss function yields our near-optimal statistical guarantees while preserving nearly linear runtime.

We summarize and compare the guarantees of our algorithms compared to existing algorithms in Table 1.

**Faster algorithm for the agnostic setting.** We also consider the agnostic setting where the data-generating distribution $H$ may not be in $\mathcal{H}$. We develop fast algorithms for this setting in the high privacy regime. Our methods extend the two-stage framework developed for simple hypothesis selection and achieve improved computational efficiency while maintaining strong statistical guarantees.

Assume $H$ is close to $H_{i^*}$ for some $i^*$. The algorithm

for simple hypothesis selection does not directly apply in this setting, as there might be many good hypotheses that are close to $H$. So no single hypothesis—including $H_{i^*}$—consistently outperforms all other hypotheses, which is the crucial structure our simple hypothesis selection algorithm utilizes. To address this challenge, we introduce an additional verification step based on the BelowThreshold mechanism, which privately checks whether all subsampled hypotheses are sufficiently poor. Then we are in a win-win situation. If this verification step fails, it certifies that a good hypothesis is in the subsampled set, in which case we can just output this hypothesis. If this verification step succeeds, then all hypotheses in the subsampled set are outperformed by $H_{i^*}$ and we can proceed to the AboveThreshold mechanism to identify a good hypothesis that outperforms all subsampled hypotheses.

**Lower bounds.** We establish a lower bound (Theorem 4.3) showing that $\Omega\big(\frac{\log m}{\Delta_\varepsilon(\mathcal{H})}\big)$ samples are needed in the worst case. This demonstrates the necessity of the dependence on $\log m$ for some families of hypotheses, and complements the existing instance-specific lower bound $\Omega\big(\frac{1}{\Delta_\varepsilon(\mathcal{H})}\big)$ in the binary case.[1]

**Fast algorithms for private universal estimation.** Finally, we show that our fast hypothesis selection algorithms lead to improved differentially private methods for universal statistical estimation in low-dimensional settings. Compared to existing approaches such as the $T$-mechanism, our algorithms achieve polynomial improvements in runtime under natural assumptions, while preserving similar near-optimal statistical accuracy.

### 1.2. Other related work

Hypothesis selection has been studied in various settings. Without considering privacy, a large body of works has

---

[1] We also remark that an instance-optimal lower bound for general $m$ would be the ultimate goal, but such a lower bound is not even known in the non-private case, which is an interesting open question.

focused on designing faster algorithms under various assumptions (Daskalakis & Kamath, 2014; Acharya et al., 2014; 2018; Aliakbarpour et al., 2023; Aamand et al., 2025). The setting where the output does not necessarily belong to the hypothesis set, called the improper setting, has also been investigated (Bousquet et al., 2019; 2022; Aamand et al., 2025). In addition to the central model of differential privacy considered in this paper, hypothesis selection has also been studied in the local model of differential privacy (Duchi & Rogers, 2019; Gopi et al., 2020; Pour et al., 2024; Kamath et al., 2025).

## 2. Problem Setting

**Hypothesis Selection**  In the problem of hypothesis selection, we have a set $\mathcal{H} := \{H_1, H_2, \ldots, H_m\}$ of $m$ publicly known distributions over a domain $\mathcal{X}$. Given a dataset of $n$ i.i.d. samples $x_{1:n} = \{x_1, x_2, ..., x_n\}$ drawn from $H_{i^*} \in \mathcal{H}$ for some $i^* \in [m]$, the goal is to identify $i^*$ with as few samples as possible. This is a well studied problem in statistics (Yatracos, 1985; Devroye & Lugosi, 1996; 1997).

In this work, we are mainly interested in private algorithms for hypothesis selection. We use the standard notion of differential privacy (Dwork et al., 2006; Dwork & Roth, 2014).

**Definition 2.1** (Differential privacy)**.**  A randomized algorithm $\mathcal{A} : \mathcal{X}^* \to \mathcal{R}$ is $\varepsilon$-differentially private ($\varepsilon$-DP) if for all $n \geq 1$, all neighboring datasets $D, D' \in \mathcal{X}^n$ that differ in at most a single datapoint, and all events $S \subseteq \mathcal{R}$, we have

$$\Pr\left[\mathcal{A}(D) \in S\right] \leq e^\varepsilon \cdot \Pr\left[\mathcal{A}(D') \in S\right] .$$

**Private Hypothesis Selection**  In the private version of hypothesis selection, the goal is to recover $i^*$ given the input dataset $x_{1:n}$ under the constraint of $\varepsilon$-DP. The following definition formalizes this problem.

**Definition 2.2** (Private hypothesis selection)**.**  Let $\mathcal{H} := \{H_1, H_2, \ldots, H_m\}$ be a set of distributions. Let $\varepsilon, \beta > 0$. We say an $\varepsilon$-DP algorithm $\mathcal{A}$ solves the hypothesis selection problem on $\mathcal{H}$ with sample complexity $\mathrm{SC}^{\mathcal{H}}_{\varepsilon,\beta}(\mathcal{A})$ if, for any $i^* \in [m]$ and any $n \geq \mathrm{SC}^{\mathcal{H}}_{\varepsilon,\beta}(\mathcal{A})$, given $n$ samples from $H_{i^*}$, the algorithm outputs $i^*$ with probability at least $1 - \beta$. We define the sample complexity of the private hypothesis selection problem on $\mathcal{H}$ to be

$$\mathrm{SC}^{\mathcal{H}}_{\varepsilon,\beta} := \min_{\mathcal{A} \text{ is } \varepsilon\text{-DP}} \mathrm{SC}^{\mathcal{H}}_{\varepsilon,\beta}(\mathcal{A}) .$$

This problem has received increased attention recently and its optimal sample complexities are known for the binary setting ($m = 2$). We will use the following definition introduced by (Asi et al., 2024) that is useful for characterizing the sample complexity of private hypothesis selection.

**Definition 2.3** ($D_\varepsilon$-divergence)**.**  Let $\varepsilon > 0$. Given two distributions $P, Q$ over the same domain, we define their $D_\varepsilon$-divergence to be

$$D_\varepsilon(P, Q) := \int (P(x) - Q(x)) \left[\log \frac{P(x)}{Q(x)}\right]^\varepsilon_{-\varepsilon} dx . \quad (1)$$

We define the $D_\varepsilon$-divergence separation of a set $\mathcal{H}$ of distributions to be

$$\Delta_\varepsilon(\mathcal{H}) := \min_{P,Q \in \mathcal{H} : P \neq Q} D_\varepsilon(P, Q) . \quad (2)$$

(Canonne et al., 2019) characterized the optimal sample complexity for $\varepsilon$-differential-privately distinguishing $P$ and $Q$. For $\varepsilon \leq O(1)$, following the divergence-based analysis of (Asi et al., 2024), the optimal sample complexity is $\Theta(1/D_\varepsilon(P, Q))$.

(McMillan et al., 2022) gave an optimal DP algorithm (based on (Canonne et al., 2019)) that first adds Laplace noise to the following (shifted) clamped log-likelihood

$$\frac{2}{n} \sum_{i=1}^n \left[\log \frac{P(x_i)}{Q(x_i)}\right]^\varepsilon_{-\varepsilon}$$
$$- \mathbb{E}_P \left[\log \frac{P(X)}{Q(X)}\right]^\varepsilon_{-\varepsilon} + \mathbb{E}_Q \left[\log \frac{Q(X)}{P(X)}\right]^\varepsilon_{-\varepsilon} .$$

Then it outputs $P$ if this (noisy) quantity is above zero, and $Q$ otherwise. (Asi et al., 2024) gave an algorithm, called private T-mechanism, that extends the above idea to the general case by instantiating the exponential mechanism with the following score function

$$f(H_i) := \min_{j \in [m] \setminus \{i\}} \left\{ \frac{2}{n} \sum_{k=1}^n \left[\log \frac{H_i(x_k)}{H_j(x_k)}\right]^\varepsilon_{-\varepsilon} \right.$$
$$\left. - \mathbb{E}_{H_i} \left[\log \frac{H_i(X)}{H_j(X)}\right]^\varepsilon_{-\varepsilon} + \mathbb{E}_{H_j} \left[\log \frac{H_j(X)}{H_i(X)}\right]^\varepsilon_{-\varepsilon} \right\} .$$

The sample complexity of their private T-mechanism is $O\left(\frac{\log m}{\Delta_\varepsilon(\mathcal{H})}\right)$.

**Notation**  The following notation is used throughout the paper. We write $f \lesssim g$ to denote $f \leq O(g)$ and $f \gtrsim g$ to denote $f \geq \Omega(g)$. We write $\tilde{O}(f)$ to denote $O(f \cdot \mathrm{polylog}(f))$ and $\tilde{\Omega}(f)$ to denote $\Omega(f / \mathrm{polylog}(f))$. We define $[x]^u_\ell := \max\{\ell, \min\{x, u\}\}$. For a matrix $M$, we define $\|M\|_{\max} := \max_{i,j} |M_{ij}|$.

## 3. Private Tournament

In this section, we introduce the *private tournament* problem, of which private simple hypothesis selection is a special case. Specifically, there are $m$ players $P_1, P_2, ..., P_m$ participating in a round-robin tournament. The results of the

tournament are recorded in an $m$-by-$m$ matrix $M$. We make the following assumptions on $M$.

1. The matrix $M$ is anti-symmetric, i.e. for all $i, j \in [m]$, $M(i,j) = -M(j,i)$. In particular, $M(i,j)$ records the score by which $P_i$ loses to $P_j$, or equivalently, the score by which $P_j$ defeats $P_i$.

2. The matrix $M$ is obtained from a sensitive dataset and changing one datapoint could change every entry of $M$ by 1. Formally, for $M$ and $M'$ obtained from neighboring datasets, we have $\|M - M'\|_{\max} \leq 1$.

3. We can access the matrix $M$ by querying $M(i,j)$ for each $i, j \in [m]$.

Our goal is to design an $\varepsilon$-DP algorithm that given query access to $M$ outputs the winner of the tournament. The utility requirement is that, when there exists an absolute winner $P_{i^*}$ such that $M(i, i^*) \geq \Delta$ for every $i \neq i^*$, we need to output $P_{i^*}$ (with high probability). The margin $\Delta$ can be regarded as the signal strength. Intuitively, larger $\Delta$ should make it easier to identify $P_{i^*}$. We measure the computational efficiency of an algorithm by its query complexity, i.e. the number of entries of $M$ it queries.

Our starting point is the following Algorithm 1 (of which the idea appeared in previous works, e.g., (Aden-Ali et al., 2021; Asi et al., 2024)). For each player $i \in [m]$, compute $\ell_i := \max_{j \in [m] \setminus \{i\}} M(i, j)$. Then sample a player $i \in [m]$ with probability $\propto e^{-\varepsilon \cdot \ell_i/2}$. Following a standard analysis of exponential mechanisms, this algorithm is $\varepsilon$-DP; and when there exists an $i^*$ such that $M(i, i^*) \geq \Omega(\log(m/\beta)/\varepsilon)$ for every $i \neq i^*$, this algorithm outputs $i^*$ with probability at least $1 - \beta$. The main drawback of this algorithm is that it needs to query all $\Theta(m^2)$ entries of the matrix $M$.

---

**Algorithm 1** Basic Algorithm

For $i \in [m]$, $\ell_i \leftarrow \max_{j:j \neq i} M(i,j)$

Sample a player $i \in [m]$ with probability $\propto e^{-\varepsilon \cdot \ell_i/2}$

---

In Section 3.1, we will present a novel $\varepsilon$-DP algorithm that makes only $O(m)$ queries while achieving the same utility as Algorithm 1. This algorithm suffices for the application to simple hypothesis selection. However, this algorithm breaks down beyond simple hypothesis selection, as it exploits heavily the assumption that there exists an absolute winner. In Section 3.2 and Section 3.3, we will present algorithms that make slightly more than $O(m)$ queries, but are robust enough to apply to non-simple hypothesis selection.

### 3.1. Private Linear Scan Algorithm

In this section, we present an $\varepsilon$-DP algorithm that makes $O(m)$ queries and outputs $i^*$ with high probability given

$M(i, i^*) \geq \tilde{\Omega}(\log(m)/\varepsilon)$ for all $i \neq i^*$. The algorithm has two stages: first, we do a linear scan over all players in an arbitrary order to find a non-private winner $c$; second, we use $c$ to carefully design a score function and run an exponential mechanism over all players.

---

**Algorithm 2** Private Linear Scan

**Parameters**: Privacy parameter $\varepsilon$

*// Find a Good Player*
$c \leftarrow 1$
**For** $i = 2, 3, \ldots, m$,
    **If** $M(c, i) > 0$ **then** $c \leftarrow i$

*// Exponential Mechanism*

$$\text{score}(i) \leftarrow \begin{cases} \max\left\{0, \min_{j:j \neq c} M(j, c)\right\}, & i = c \\ 0, & i \neq c \end{cases} \quad (3)$$

Sample a player $i \in [m]$ with probability $\propto e^{\frac{\varepsilon}{2} \cdot \text{score}(i)}$

---

**Theorem 3.1** (Private linear scan algorithm). *For every privacy parameter $\varepsilon$, Algorithm 2 satisfies the following.*

- *Privacy: The algorithm is $\varepsilon$-DP.*

- *Utility: If there exists an $i^* \in [m]$ such that $M(i, i^*) > 2\log(m/\beta)/\varepsilon$ for all $i \in [m] \setminus \{i^*\}$, then the algorithm outputs $i^*$ with probability at least $1 - \beta$.*

- *Efficiency: The algorithm queries at most $2m$ entries of $M$.*

*Proof. Privacy:* Let $M, M'$ be obtained from neighboring datasets. By Lemma A.1, it suffices to show the sensitivity of score is bounded by 1. There are two cases. Case 1: $c = c'$. This case is straightforward by Lemma A.5. Case 2: $c \neq c'$. In this case, we will first show $\text{score}(c) \leq 1$. Assume for contradiction $\text{score}(c) > 1$. Then it must be $\min_{i:i \neq c} M(i, c) > 1$. Since $\|M - M'\|_{\max} \leq 1$, we have $\min_{i:i \neq c} M'(i, c) > 0$. That is, the same player $c$ will also be the final winner in stage 1 when we move to the neighboring dataset, contradicting $c \neq c'$. Thus, $\text{score}(c) \leq 1$. A symmetric argument will show $\text{score}'(c') \leq 1$. Once given $0 \leq \text{score}(c), \text{score}'(c') \leq 1$, it is easy to see the sensitivity is at most 1.

*Utility:* Assume there exists an $i^* \in [m]$ with $\Delta := \min_{i:i \neq i^*} M(i, i^*) > 0$. We want to show stage 2 outputs $i^*$ w.h.p. It is easy to see stage 1 will return $i^*$ deterministically, i.e., $c = i^*$. Then $\text{score}(i^*) = \Delta$ and $\text{score}(i) = 0$ for every other $i \neq i^*$. Therefore, by Lemma A.1, as long as $\Delta > \frac{2\log(m/\beta)}{\varepsilon}$, the exponential mechanism will return $i^*$ with probability at least $1 - \beta$.

*Efficiency:* In the first stage, the algorithm makes $m - 1$ queries. In the second stage, the algorithm makes at most another $m - 1$ queries. So this algorithm makes at most $2m$ queries. □

## 3.2. Sample-and-Prune Tournament Algorithm

In order to be able to apply to non-simple hypothesis selection, we would need the utility guarantee of a tournament algorithm to be in the following form. If there exists a player $i^*$ such that $\max_j M(i^*, j) \leq B$ for some $B \geq 0$, then the algorithm should find some player $i$, not necessarily $i^*$, satisfying $\max_j M(i, j) \leq O(B)$ with high probability. In this section and the following section, we present algorithms that can be easily adapted to apply to non-simple hypothesis selection. For the purpose of readability, we defer the adaption to Appendix C.

Now we present our new algorithm that makes $\widetilde{O}(m^{1.5})$ queries and, when there exists an $i^*$ with $M(i, i^*) \geq \widetilde{\Omega}(\log(m)/\varepsilon)$ for all $i \neq i^*$, outputs $i^*$ with high probability. Similar to Algorithm 2, the algorithm leverages a similar two-stage design: first, it samples a subset of the players to identify a strong player; then, it exploits this player to effectively run exponential mechanism over the set of players that are not defeated by the strong player.

**Theorem 3.2** (Sample-and-Prune algorithm). *For every privacy parameter $\varepsilon$ and failure probability $\beta$, Algorithm 3 satisfies the following.*

- *Privacy: The algorithm is $\varepsilon$-DP.*

- *Utility: If there exists an $i^* \in [m]$ such that $M(i, i^*) \geq 480 \log(4m/\beta)/\varepsilon$ for all $i \in [m] \setminus \{i^*\}$, then the algorithm outputs $i^*$ with probability at least $1 - \beta$.*

- *Efficiency: The algorithm queries $O(m^{1.5} \log m)$ entries of $M$ in expectation.*

Next we present the key intuition underlying Algorithm 3. Our starting point is the trivial observation that player $i^*$ defeats everyone else by at least $\Delta = \Theta(\log(m/\beta)/\varepsilon)$. We quantify the goodness of a player by the number of other players it defeats by a margin of at least $\Delta$. In the first stage of our algorithm, we privately identify a sufficiently strong player $\tilde{c}$ that beats many players. A crucial observation is that if a player $c$ defeats a random subset of $k$ players by $\Delta$, then the number of players that $c$ does not beat by $\Delta$ is at most $\tilde{O}(m/k)$. Leveraging this insight, in the first stage we sample a random subset $S \subset [m]$ and privately find a player $c \in [m]$ that defeats all players in $S$. To achieve this while incurring minimal privacy cost, we employ the AboveThreshold mechanism.

In the second stage, given the privately identified strong player $\tilde{c}$, one might consider applying the basic algorithm

(Algorithm 1) to the set of remaining players in $[m]$ not defeated by $\tilde{c}$, i.e. the set $\{i \in [m] : M(i, \tilde{c}) < \Delta\}$. However, this approach would violate differential privacy as this set itself is sensitive. In fact, the size of this set could change by $\Omega(m)$ when we move to a neighboring dataset. To preserve privacy, we apply the exponential mechanism over the set of all players using a carefully designed loss function that avoids additional queries for players defeated by $\tilde{c}$. Specifically, if $M(i, \tilde{c}) \geq 0.8\Delta$ for a player $i \in [m]$, we set $\ell_i = M(i, \tilde{c})$. Otherwise, we will compare this player against all other players, and project their loss to bound the sensitivity, setting $\ell_i = \min\{0.8\Delta, \max_{j:j \neq i} M(i, j)\}$. Finally, the algorithm runs the exponential mechanism over $[m]$ with this loss function and returns its output.

To develop intuition for the query complexity, observe that the first stage requires $O(m \cdot k)$ queries where $k = |S|$ denotes the size of the sampled subset. In the second stage, the algorithm makes $m$ queries only for players not defeated by $\tilde{c}$, of which there are at most $\tilde{O}(m/k)$. Thus, the query complexity is roughly $mk + m^2/k$. Setting $k = \sqrt{m}$ yields an algorithm with $\tilde{O}(m^{1.5})$ queries.

---

**Algorithm 3** Sample-and-Prune Algorithm

**Parameters**: Privacy parameter $\varepsilon$, failure probability $\beta$

$$\Delta \leftarrow \frac{480 \log(4m/\beta)}{\varepsilon}$$

*// Find a Good Player*
$p \leftarrow 1/\sqrt{m}$
$S \leftarrow$ Subsample $[m]$ with probability $p$
**For** $c \in [m]$,

$$\text{score}(c) \leftarrow \begin{cases} \min_{i \in S \setminus \{c\}} M(i, c), & S \setminus \{c\} \neq \emptyset \\ \Delta, & \text{otherwise} \end{cases} \quad (4)$$

$\tilde{c} \leftarrow \text{AboveThreshold}(\{\text{score}(c)\}, \varepsilon/2, 0.9\Delta)$
**If** $\tilde{c} = \perp$ **then abort**

*// Exponential Mechanism*
**For** $i \in [m]$,

$$\ell_i \leftarrow \begin{cases} \left[\max_{j:j \neq i} M(i, j)\right]_{-\infty}^{0.8\Delta}, & M(i, \tilde{c}) < 0.8\Delta \\ M(i, \tilde{c}), & M(i, \tilde{c}) \geq 0.8\Delta \end{cases} \quad (5)$$

Sample a player $i \in [m]$ with probability $\propto e^{-\frac{\varepsilon}{4} \cdot \ell_i}$

---

*Remark* 3.3. There is more than one method for the subsampling step in Algorithm 3. For example, instead of putting each member of $[m]$ in $S$ with probability $1/\sqrt{m}$ independently, we could also sample uniformly at random elements from $[m]$ repeatedly for $\sqrt{m}$ times with or without replacement. This is because we are sampling from a fixed

set $[m]$ that does not depend on the underlying dataset. As we will see in Section 3.3, this will not be the case if we want to generalize the algorithm to multiple rounds.

Now we proceed to prove Theorem 3.2. The proof will follow directly from Lemma 3.5 (Privacy), Lemma 3.6 (Utility), and Lemma 3.7 (Efficiency).

We begin by proving privacy. The following claim is the key step in the privacy analysis, showing that the sensitivity of our score function (5) in Algorithm 3 is bounded by 1.

**Claim 3.4.** *The sensitivity of the function in Equation* (5) *is at most* 1.

*Proof.* Let $M, M'$ be the matrices obtained from neighboring datasets. We have $\|M - M'\|_{\max} \leq 1$. There are four cases to consider, but we only need to consider two cases by symmetry.

Case 1: $M(i, \tilde{c}), M'(i, \tilde{c}) < 0.8\Delta$. The sensitivity bound directly follows from Lemma A.5.

Case 2: $M(i, \tilde{c}) < 0.8\Delta \leq M'(i, \tilde{c})$. Observe

$$M(i, \tilde{c}) \leq \ell_i \leq 0.8\Delta \leq M'(i, \tilde{c}) = \ell'_i.$$

Then, $|\ell_i - \ell'_i| \leq |M(i, \tilde{c}) - M'(i, \tilde{c})| \leq 1$. □

**Lemma 3.5** (Privacy). *Algorithm 3 is $\varepsilon$-DP.*

*Proof.* To prove privacy, we analyze the two main stages of the algorithm separately, prove that each stage is $\varepsilon/2$-DP, and then use composition to argue that the final algorithm is $\varepsilon$-DP. For the first step, note that Lemma A.5 implies that the sensitivity of the function in Equation (4) is at most 1. Therefore, the guarantees of the AboveThreshold algorithm (Lemma A.2) imply that $\tilde{c}$ is $\varepsilon/2$-DP. For the second stage, Claim 3.4 implies that the sensitivity of the function in Equation (5) is also at most 1. Therefore its output is $\varepsilon/2$-DP by the guarantees of the exponential mechanism (Lemma A.1). Finally, using DP composition (Lemma A.4), Algorithm 3 is $\varepsilon$-DP. □

Next, we move to prove our claim about utility. The key idea is the loss function (5) defined in the second stage maintains a separation in the values between $\ell_{i^*}$ and $\ell_i$ for $i \neq i^*$. Therefore, the exponential mechanism picks $i^*$ with high probability.

**Lemma 3.6** (Utility). *Suppose there exists an $i^* \in [m]$ such that $M(i, i^*) \geq 480\log(4m/\beta)/\varepsilon$ for all $i \in [m] \setminus \{i^*\}$. Then Algorithm 3 outputs $i^*$ with probability at least $1 - \beta$.*

*Proof.* By assumption, for all possible subsets $S$,

$$\text{score}(i^*) \geq \Delta \geq 0.9\Delta + \frac{8\log(4m/\beta)}{\varepsilon/2}.$$

Then by the utility guarantee of AboveThreshold (Lemma A.2), AboveThreshold returns $\tilde{c} \neq \perp$ with probability at least $1 - \beta/2$. Conditioned on $\tilde{c} \neq \perp$, the definition of the loss $\ell_i$ (see Eq. (5)) implies that $\ell_{i^*} \leq -\Delta$ and $\ell_i \geq 0.8\Delta$ for all $i \neq i^*$. Therefore, by Lemma A.1, the exponential mechanism outputs $i^*$ with probability at least $1 - \beta/2$. By a union bound, Algorithm 3 outputs $i^*$ with probability at least $1 - \beta$. □

Next, we prove an upper bound on the number of queries that the algorithm makes. The main observation is that the number of players that are not eliminated by $\tilde{c}$ is small, hence the second stage of the algorithm requires few queries.

**Lemma 3.7** (Efficiency). *Algorithm 3 queries at most $O(m^{1.5} \log m)$ entries of $M$ in expectation.*

*Proof.* First, note that the first stage of the algorithm makes at most $m^{1.5}$ queries in expectation: indeed, in the worst case, for each player $c \in [m]$ the algorithm makes $|S|$ queries to calculate $\text{score}(c)$. Thus the expected number of queries in the first stage is

$$\mathbb{E}[m \cdot |S|] = m \cdot mp = m^{1.5}.$$

Next, we show that the second stage of the algorithm also queries at most $O(m^{1.5} \log m)$ entries. The key idea is to show that most players are eliminated by $\tilde{c}$, that is $M(i, \tilde{c}) \geq 0.8\Delta$, and therefore the calculation of their loss $\ell_i$ requires reading a single entry. Therefore, we define the set of remaining players which were not eliminated by $\tilde{c}$:

$$S_{\text{rem}} := \{i \in [m] : M(i, \tilde{c}) < 0.8\Delta\},$$

and argue that the size of this set is small. It is immediate that the number of queries that the algorithm makes in the second stage is at most

$$m + |S_{\text{rem}}| \cdot m.$$

It remains to prove an upper bound on $|S_{\text{rem}}|$. Let $B$ denote the (bad) event that $\tilde{c} \neq \perp$ but $\text{score}(\tilde{c}) < 0.8\Delta$. As $\Delta \geq 480\log(2m)/\varepsilon$, we have $\Pr(B) \leq 1/m^2$, by Lemma A.2. Conditioned on the event that $B$ does not happen, for every $k \in \mathbb{N}$, the event $|S_{\text{rem}}| \geq k$ implies the following event: there exists a $c \in [m]$ such that the set $\{i \in [m] : M(i, c) < 0.8\Delta\}$ has size at least $k$ and none of its element is sampled. Thus,

$$\Pr(|S_{\text{rem}}| \geq k \,|\, \bar{B}) \leq \min\{m(1-p)^k, 1\}.$$

Let $K := \max\{k \in \mathbb{Z}_{\geq 0} : m(1-p)^k \geq 1\}$. Note

$K = \left\lfloor \frac{\ln m}{\ln 1/(1-p)} \right\rfloor$. Then

$$
\begin{aligned}
\mathbb{E}\left[|S_{\mathsf{rem}}| \,\big|\, \bar{B}\right] &= \sum_{k=1}^{\infty} \Pr\left(|S_{\mathsf{rem}}| \geq k \,\big|\, \bar{B}\right) \\
&\leq K + 1 + \frac{m(1-p)^{K+1}}{p} \\
&\leq 2 + \frac{1 + \ln m}{p} \ .
\end{aligned}
$$

Thus,

$$
\begin{aligned}
\mathbb{E}\,|S_{\mathsf{rem}}| &= \mathbb{E}\left[|S_{\mathsf{rem}}| \,\big|\, \bar{B}\right]\Pr(\bar{B}) + \mathbb{E}\left[|S_{\mathsf{rem}}| \,\big|\, B\right]\Pr(B) \\
&\leq \mathbb{E}[|S_{\mathsf{rem}}| \,|\, \bar{B}] + m^2 \cdot \Pr(B) \\
&\leq 3 + \frac{1 + \ln m}{p} \ .
\end{aligned}
$$

Finally, plugging $p = 1/\sqrt{m}$, we have that the expected number of queries made in the second stage is at most

$$
m + \mathbb{E}[S_{\mathsf{rem}}] \cdot m \leq O\left(m^{1.5}\ln m\right) .
$$

$\square$

### 3.3. Multi-Round Sample-and-Prune Algorithm

In Section 3.2, we introduced a Sample-and-Prune algorithm with query complexity $\widetilde{O}(m^{1.5})$. In this section, we obtain even stronger results by extending the sampling stage of our algorithm to work multiple rounds: instead of finding a single strong player, the key idea is to repetitively for $T$ rounds find a new strong player given the previously identified strong players. We show that this strategy yields a $T$-round algorithm with query complexity $\widetilde{O}((T+1)\cdot m^{1+\frac{1}{T+1}})$ for every non-negative integer $T$.

**Theorem 3.8** (Multi-round algorithm)**.** *For every non-negative integer $T$, privacy parameter $\varepsilon$, and failure probability $\beta$, Algorithm 4 satisfies the following.*

- *Privacy: The algorithm is $\varepsilon$-DP.*

- *Utility: If there exist an $i^* \in [m]$ and a sufficiently large constant $C$ such that $M(i, i^*) \geq \frac{CT^2 \log(Tm/\beta)}{\varepsilon}$ for all $i \in [m] \setminus \{i^*\}$, then the algorithm outputs $i^*$ with probability at least $1 - \beta$.*

- *Efficiency: The algorithm in expectation queries at most $O((T+1)\, m^{1+\frac{1}{T+1}} \log m)$ entries of $M$.*

Due to space limitations, we defer the proof of Theorem 3.8 to Appendix B.

*Remark* 3.9. Note that we can recover the results for the previous two algorithms as a special case of this result. Indeed, setting $T = 0$ recovers the basic exponential mechanism with $O(m^2)$ query complexity (Algorithm 1), and

setting $T = 1$ recovers the Sample-and-Prune algorithm with $\widetilde{O}(m^{1.5})$ query complexity (Algorithm 3). Moreover, setting larger values $T > 1$ allows us to obtain a nice trade-off between the margin $\min_{i:\, i \neq i^*} M(i, i^*)$ and the query complexity. In particular, setting $T = \log \log m$, Algorithm 4 obtains near-linear query complexity $m^{1+o(1)}$ while only requiring that $\min_{i:\, i \neq i^*} M(i, i^*) \geq \tilde{\Omega}(\log(m/\beta)/\varepsilon)$. This requirement on the margin is not strict for our application in hypothesis selection as we will see in the next section.

Next, we present the main idea of Algorithm 4. Recall that in Algorithm 3, we first privately identify a single strong player and then apply the exponential mechanism essentially over the set of players not defeated by this candidate. A natural extension of this approach is to identify multiple strong players in the first stage. For clarity, we describe how to privately identify two strong players; the extension to $T$ strong players follows naturally.

Suppose we have already identified the first strong player $\tilde{c}_1$. Let $R$ denote the set of players that are not defeated by $\tilde{c}_1$. A natural candidate for a second strong player is one that defeats many players in $R$. One might attempt to repeat the procedure from Algorithm 3 by subsampling a set $S \subset R$ and applying the AboveThreshold mechanism to select a player from $[m] \setminus \{\tilde{c}_1\}$ that defeats all players in $S$. Unfortunately, this approach fails to preserve differential privacy, since both $R$ and its subsampled version $S$ may change arbitrarily between neighboring datasets.

A standard approach in the differential privacy literature would be to privatize the subsampling step itself so that $S$ becomes private. However, this would incur an unacceptably large privacy cost. Instead, our method departs from this conventional approach and is based on the following key observation. Let $R$ and $R'$ be obtained from neighboring datasets. If a player $i$ is in $R$ but not in $R'$, then $i$ must be nearly defeated by $\tilde{c}_1$, since

$$
|M(i, \tilde{c}_1) - M'(i, \tilde{c}_1)| \leq 1
$$

This stability property implies that, with a carefully designed score function, we can safely use a randomly chosen subset of $R$ to privately select our second strong player $\tilde{c}_2$, even though $R, R'$ could differ arbitrarily.

Once a small set of strong players has been privately identified, the second stage of the algorithm proceeds analogously to Algorithm 3.

To develop intuition for query complexity, observe that finding the first strong player makes $O(p_1 m^2)$ queries and the second strong player makes $\tilde{O}(p_2/p_1 m)$. In the second stage, the algorithm makes $m$ queries only for players not defeated by both $\tilde{c}_1$ and $\tilde{c}_2$, of which there are at most $\tilde{O}(1/p_2)$. Thus, the query complexity is roughly

$p_1 m^2 + p_2/p_1 m + m/p_2$, of which the minimum is $m^{4/3}$ by setting $p_1 = m^{-2/3}$ and $p_2 = m^{-1/3}$. Repeating this for $T$ rounds results in even better query complexity, as we show in the proof.

We present the full details in Algorithm 4. To simplify our algorithmic notation, we introduce the following definition: given $i \in [m]$ and $G \subseteq [m]$, define

$$\text{loss}(i, G) := \begin{cases} \max_{j \in G} M(i,j), & G \neq \emptyset \\ -\infty, & G = \emptyset \end{cases}.$$

## 4. Private Hypothesis Selection via Private Tournaments

In this section, we apply our private tournament algorithms to the private hypothesis selection problem.

**Theorem 4.1.** *Consider the simple hypothesis selection problem for a set $\mathcal{H} := \{H_1, \ldots, H_m\}$ of distributions. There is an $\varepsilon$-DP algorithm that solves it with sample complexity $O\left(\frac{\log(m/\beta)}{\Delta_\varepsilon(\mathcal{H})}\right)$ and runs in time $O(mn)$ on $n$ samples.*

*Proof.* Given a dataset $\{x_1, \ldots, x_n\}$, we define the matrix $M$ as follows,

$$M(i,j) := \frac{1}{2\varepsilon} \sum_{k=1}^n \left[ \log \frac{H_j(x_k)}{H_i(x_k)} \right]_{-\varepsilon}^{\varepsilon}$$
$$+ \frac{n}{4\varepsilon} \mathbb{E}_{H_i} \left[ \log \frac{H_i(X)}{H_j(X)} \right]_{-\varepsilon}^{\varepsilon} - \frac{n}{4\varepsilon} \mathbb{E}_{H_j} \left[ \log \frac{H_j(X)}{H_i(X)} \right]_{-\varepsilon}^{\varepsilon}.$$

It is easy to see $M$ is anti-symmetric and each entry of $M$ has sensitivity at most 1.

Assume $x_1, \ldots, x_n$ are independently drawn from $H_{i^*}$ for some $i^* \in [m]$. Observe that for all $i \in [m]$,

$$\mathbb{E}\, M(i, i^*) = \frac{n}{4\varepsilon} D_\varepsilon(H_{i^*}, H_i).$$

Applying the Bernstein concentration inequality and a union bound, for $n \gtrsim \frac{\log(m/\beta)}{\min_{j \neq i^*} D_\varepsilon(H_{i^*}, H_j)}$, we have with probability at least $1 - \beta$ that for all $i \in [m]$,

$$0.9\, \mathbb{E}\, M(i, i^*) \leq M(i, i^*) \leq 1.1\, \mathbb{E}\, M(i, i^*).$$

Running Algorithm 2 on the above matrix $M$, we would obtain $i^*$ as the output with probability $1 - \beta$. As the algorithm queries $O(m)$ entries of $M$, its running time is $O(mn)$. □

### 4.1. Beyond Simple Hypothesis Selection

In this section, we consider the case where the true distribution does not necessarily belong to the hypothesis set.

---

**Algorithm 4** Multi-Round Algorithm

**Parameters**: Privacy parameter $\varepsilon$, failure probability $\beta$, number of rounds $T$

$\Delta \leftarrow \frac{CT^2 \log(Tm/\beta)}{\varepsilon}$ for a sufficiently large constant $C$
$\varepsilon_t \leftarrow \frac{\varepsilon}{2T}$ for $t \in [T]$, and $\varepsilon_{\text{EM}} \leftarrow \frac{\varepsilon}{2}$
$\beta_t \leftarrow \frac{\min\{\beta/2, 1/m^2\}}{T}$ for $t \in [T]$, and $\beta_{\text{EM}} \leftarrow \frac{\beta}{2}$
$p_t \leftarrow m^{-1+\frac{t}{T+1}}$ for $t \in [T]$

*// Find Good Players*
Initialize $G_0 \leftarrow \emptyset$ and $\Delta_0 \leftarrow \Delta$
**For** $t = 1, 2, \ldots, T$:
    $R_t \leftarrow \{i \in [m] : \text{loss}(i, G_{t-1}) < \Delta_{t-1}\}$
    $S_t \leftarrow$ Subsample $R_t$ with probability $p_t$
    **For** $c \in [m] \setminus G_{t-1}$:
        **If** $S_t \setminus \{c\}$ is nonempty **then**

$$\text{score}_t(c) \leftarrow \min_{i \in S_t \setminus \{c\}} \left[ \text{loss}(i, G_{t-1} \cup \{c\}) \right]_{-\infty}^{\Delta_{t-1}} \tag{6}$$

        **Else** $\text{score}_t(c) \leftarrow \Delta_{t-1}$
    $\tau_t \leftarrow \Delta_{t-1} - \frac{8 \log(2m/\beta_t)}{\varepsilon_t}$
    $\tilde{c}_t \leftarrow \text{AboveThreshold}(\{\text{score}_t(c)\}, \varepsilon_t, \tau_t)$
    **If** $\tilde{c}_t = \perp$ **then**
        $G_T \leftarrow G_{t-1} \cup \{\perp\}$
        $\Delta_T \leftarrow \Delta_{t-1}$
        **break**
    **Else**
        $G_t \leftarrow G_{t-1} \cup \{\tilde{c}_t\}$
        $\Delta_t \leftarrow \tau_t - \frac{8 \log(2m/\beta_t)}{\varepsilon_t}$

*// Exponential Mechanism*
**If** $G_T = \{\perp\}$ **then** abort
**Else if** $\perp \in G_T$ **then** $\ell_i \leftarrow \text{loss}(i, G_T \setminus \{\perp\})$ for $i \in [m]$
**Else if** $\perp \notin G_T$ **then** for $i \in [m]$

$$\ell_i \leftarrow \begin{cases} \left[ \max_{j:j \neq i} M(i,j) \right]_{-\infty}^{\Delta_T}, & \text{loss}(i, G_T) < \Delta_T \\ \text{loss}(i, G_T), & \text{loss}(i, G_T) \geq \Delta_T \end{cases} \tag{7}$$

Sample a player $i \in [m]$ with probability $\propto e^{-\varepsilon_{\text{EM}} \cdot \ell_i / 2}$

---

**Theorem 4.2.** *Let $\mathcal{H} := \{H_1, \ldots, H_m\}$ be a set of distributions. Let $P$ be a distribution such that $\|P - H_{i^*}\|_{\text{TV}} \leq \eta$ for some $i^* \in [m]$. For every positive integer $T$, privacy parameter $\varepsilon \leq O(\eta)$, and failure probability $\beta$, there is an $\varepsilon$-DP algorithm that given $O\left(\frac{T^2 \log(Tm/\beta)}{\varepsilon\eta}\right)$ samples from $P$, outputs an $i \in [m]$ satisfying $D_\varepsilon(P, H_i) \leq O(\varepsilon\eta)$ with probability at least $1 - \beta$. Moreover, the algorithm has expected running time $O(T \cdot nm^{1+\frac{1}{T+1}} \log m)$ on $n$ samples.*

Due to space limitations, we defer the proof to Appendix C.

## 4.2. Lower Bound

We show that the sample complexity of the private T-mechanism introduced by (Asi et al., 2024) as well as the algorithms we introduce in this paper are optimal on some instances.

**Theorem 4.3.** *There exists a set $\mathcal{H}$ of $m$ distributions such that $\mathrm{SC}_{\varepsilon,\beta}^{\mathcal{H}} \geq \Omega\left(\frac{\log(m/\beta)}{\Delta_\varepsilon(\mathcal{H})}\right)$.*

*Proof.* We will first prove for any set $\mathcal{H}$ of $m$ distributions,

$$\mathrm{SC}_{\varepsilon,\beta}^{\mathcal{H}} \geq \Omega\left(\frac{\log(m/\beta)}{\varepsilon}\right) .$$

Let $\mathcal{A}$ be an $\varepsilon$-DP algorithm that solves the hypothesis selection problem of $\mathcal{H}$ with sample complexity $n$ and failure probability $\beta$. For each $i \in [m]$, let $D_i$ denote a set of $n$ i.i.d. samples from $H_i$. By our utility assumption, for each $i \in [m]$,
$$\Pr(\mathcal{A}(D_i) = i) \geq 1 - \beta .$$

Fix an $i^* \in [m]$. By group privacy, for each $i \in [m]$,

$$\Pr(\mathcal{A}(D_{i^*}) = i) \geq e^{-n\varepsilon} \cdot \Pr(\mathcal{A}(D_i) = i) .$$

Putting them together,

$$\sum_{i \in [m]} \Pr(\mathcal{A}(D_{i^*}) = i) \geq (1 - \beta) + (m - 1)e^{-n\varepsilon}(1 - \beta) .$$

As a probability is at most 1, we have

$$n \geq \frac{\log\left((1-\beta)(m-1)/\beta\right)}{\varepsilon} \geq \Omega\left(\frac{\log(m/\beta)}{\varepsilon}\right) .$$

Now consider any set $\mathcal{H} = \{H_1, H_2, \ldots, H_m\}$ of distributions with disjoint supports. For $i \neq j$, we have

$$D_\varepsilon(H_i, H_j) = \int (H_i(x) - H_j(x)) \left[\log \frac{H_i(x)}{H_j(x)}\right]_{-\varepsilon}^{\varepsilon} = 2\varepsilon .$$

As a result, $\Delta_\varepsilon(\mathcal{H}) = 2\varepsilon$. Therefore, for any set $\mathcal{H}$ of distributions with disjoint supports,

$$\mathrm{SC}_{\varepsilon,\beta}^{\mathcal{H}} \geq \Omega\left(\frac{\log(m/\beta)}{\Delta_\varepsilon(\mathcal{H})}\right) .$$

$\square$

# 5. Faster Algorithms for Private Universal Statistical Estimation

In this section, we use our new private selection algorithms to get faster universal algorithms for private statistical estimation of any 1-dimensional statistic. Given a set $x_{1:n} = \{x_1, \ldots, x_n\}$ of samples from a distribution $P \in \mathcal{P}$

and a statistic $\theta : \mathcal{P} \to \mathbb{R}$, our goal is to design an algorithm that privately estimates $\theta(P)$ given $x_{1:n}$.

Most closely related to our work are the prior work of (McMillan et al., 2022) and (Asi et al., 2024), which also study private estimators for general statistics. However, the work of (McMillan et al., 2022) obtains optimal rates only for generalized linear models. (Asi et al., 2024) proposed the T-mechanism which obtains near-optimal error for general low-dimensional statistics. In particular, (Asi et al., 2024) characterize the rates of private statistical estimation using the following quantity:

$$\omega_\varepsilon(\delta; P, \mathcal{P}) = \sup_{P_1 \in \mathcal{P}} \left\{|\theta(P_1) - \theta(P_0)| \mid D_\varepsilon(P, P_1) \leq \delta\right\}.$$

(Asi et al., 2024) show that the T-mechanism obtains the near-optimal rate $|\hat{\theta} - \theta(P)| \leq \omega_\varepsilon(O(\log |\mathcal{P}_\eta|/n; P, \mathcal{P})$ where $\mathcal{P}_\eta$ is an $\eta$ cover of $\mathcal{P}$ for some small $\eta$. However, this algorithm can be inefficient even for low-dimensional statistics as it runs in time $m^2 \cdot n$ where $m = |\mathcal{P}_\eta|$.

In this section, we develop new faster algorithms for private estimation. Following the work (Asi et al., 2024), we assume for simplicity that $\theta(P) \in [0, 1]$ and construct an $\eta$-cover $\mathcal{P}_\eta$ of $\mathcal{P}$ in total variation distance: for all $P \in \mathcal{P}$ there is $P_0 \in \mathcal{P}_\eta$ such that $\|P - P_0\|_{TV} \leq \eta$.

Our private estimator will apply the private hypothesis selection algorithm from the previous section (the algorithm from Theorem 4.2) over the set $\mathcal{P}_\eta$. Based on the guarantees of our algorithms, we get the following result.

**Corollary 5.1.** *Let $\theta : \mathcal{P} \to [0, 1]$, $P \in \mathcal{P}$ and $X_1, \ldots, X_n \sim P$. Let $\mathcal{P}_\eta$ be an $\eta$-cover of $\mathcal{P}$, $\varepsilon \leq \eta$, and let $m = |\mathcal{P}_\eta|$. Our algorithm is $\varepsilon$-DP, has expected running time $O(nm^{1+o(1)} \log m)$, and returns $\hat{\theta}$ that with probability $1 - \beta$ has*

$$|\hat{\theta} - \theta(P)| \leq \omega_\varepsilon \left(\frac{O\left(\log(m/\beta) \log^2 \log m\right)}{n} + \varepsilon\eta; P, \mathcal{P}\right).$$

Our algorithm achieves near-optimal rates comparable to those of the $T$-mechanism, up to an extra $\log^2 \log m$ factor. In contrast, it is substantially more computationally efficient, with a runtime of approximately $n \cdot m^{1+o(1)}$, compared to $n \cdot m^2$ for the $T$-mechanism. This computational gap is significant even in one-dimensional estimation problems.

To illustrate, consider a parametric estimation setting where $\mathcal{P} = \{P_\mu : \mu \in [0, 1]\}$ and the target parameter is $\theta(P_\mu) = \mu$. Setting $\eta = 1/\sqrt{n}$ to get small error due to covering, standard regularity assumptions imply the existence of an $\eta$-cover of $\mathcal{P}$ of size $O(\sqrt{n})$. Therefore, the $T$-mechanism runs in time $O(n^2)$ whereas our mechanism runs in time $O(n^{1.5+\rho})$ for any $\rho > 0$.

## Impact Statement

This paper presents work whose goal is to advance the field of machine learning. There are many potential societal consequences of our work, none of which we feel must be specifically highlighted here.

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

## A. Useful Results in Differential Privacy

In this section, we collect several useful results from the differential privacy literature, which will be helpful throughout the paper. We begin with the exponential mechanism.

**Lemma A.1** (Exponential mechanism). *Given $\varepsilon$ and $m$ sensitivity-1 scores $a_1, a_2, \ldots, a_m$, the* exponential mechanism *samples $\hat{i} \in [m]$ with probability $P(\hat{i} = i) \propto e^{-\varepsilon \cdot a_i / 2}$. The* exponential mechanism *is $\varepsilon$-DP and outputs an index $\hat{i} \in [m]$ such that with probability at least $1 - \beta$,*

$$a_{\hat{i}} \leq \min_{i \in [m]} a_i + \frac{2 \log(m/\beta)}{\varepsilon} \, .$$

**Lemma A.2** (AboveThreshold). *Given a threshold $\tau$ and $m$ sensitivity-1 scores $a_1, a_2, \ldots, a_m$, there is an $\varepsilon$-DP algorithm* AboveThreshold$(\{a_i\}, \varepsilon, \tau)$ *that outputs either an index $i \in [m]$ or $\perp$, and satisfies the following utility guarantee with probability at least $1 - \beta$ :*

- *If it outputs $i \in [m]$, then $a_i > \tau - \frac{8 \log(2m/\beta)}{\varepsilon}$.*

- *If it outputs $\perp$, then $\max\limits_{i \in [m]} a_i < \tau + \frac{8 \log(2m/\beta)}{\varepsilon}$.*

**Lemma A.3** (BelowThreshold). *Given a threshold $\tau$ and $m$ sensitivity-1 scores $a_1, a_2, \ldots, a_m$, there is an $\varepsilon$-DP algorithm* BelowThreshold$(\{a_i\}, \varepsilon, \tau)$ *that outputs either an index $i \in [m]$ or $\perp$, and satisfies the following utility guarantee with probability at least $1 - \beta$ :*

- *If it outputs $i \in [m]$, then $a_i < \tau + \frac{8 \log(2m/\beta)}{\varepsilon}$.*

- *If it outputs $\perp$, then $\min\limits_{i \in [m]} a_i > \tau - \frac{8 \log(2m/\beta)}{\varepsilon}$.*

**Lemma A.4** (Composition). *For $i \in [k]$, let $\mathcal{A}_i : \mathcal{X}^n \times \prod_{j=1}^{i-1} \mathcal{R}_j \to \mathcal{R}_i$ be a sequence of algorithms. Suppose $\mathcal{A}_i(\cdot, y_1, \ldots, y_{i-1})$ is $\varepsilon_i$-DP for any $y_1 \in \mathcal{R}_1, \ldots, y_{i-1} \in \mathcal{R}_{i-1}$. Then the composed algorithm $\mathcal{A} : \mathcal{X}^n \to \prod_{i=1}^{k} \mathcal{R}_i$ that runs $\mathcal{A}_i$ in sequence, is $\left( \sum_{i=1}^{k} \varepsilon_i \right)$-DP.*

We will intensively use the following lemma which says that the sensitivity of every quantile of a sequence of sensitivity-1 values is also 1.

**Lemma A.5** (Quantile sensitivity). *Let $a_1, \ldots, a_m$ and $a'_1, \ldots, a'_m$ be such that $|a_i - a'_i| \leq 1$ for all $i \in [m]$. Sort $a_1, \ldots, a_m$ non-decreasingly and get $a_{i_1} \leq \cdots \leq a_{i_m}$. Similarly, sort $a'_1, \ldots, a'_m$ and get $a'_{i_1} \leq \cdots \leq a'_{i_m}$. Then $|a_{i_j} - a'_{i_j}| \leq 1$ for every $j \in [m]$.*

## B. Proofs for Section 3.3

In this section we prove Theorem 3.8, which will follow directly from Lemma B.3 (Privacy), Lemma B.4 (Utility), and Lemma B.5 (Efficiency).

**Privacy Analysis**  Similar to the privacy analysis of Algorithm 3, the key step is to show the score function $\text{score}_t$ in Equation (6) and the losses $\{\ell_i\}$ in Equation (7) have low sensitivity. The proof that $\{\ell_i\}$ have sensitivity 1 is exactly the same as in Algorithm 3. The challenge of bounding the sensitivity of $\text{score}_t$ comes from that $R_t$ and $R'_t$ (and thus $S_t$ and $S'_t$) could differ arbitrarily. Our key observation is stated in the following lemma.

**Lemma B.1.** *For every $t \in [T]$ and every $c \in [m] \setminus G_{t-1}$,*

$$\min_{i \in R_t \setminus \{c\}} \left[ \text{loss}\big(i, G_{t-1} \cup \{c\}\big) \right]_{-\infty}^{\Delta_{t-1}}$$

$$- \min_{i \in R'_t \setminus \{c\}} \left[ \text{loss}'\big(i, G_{t-1} \cup \{c\}\big) \right]_{-\infty}^{\Delta_{t-1}} \in [-1, +1]$$

*Proof.* Fix an arbitrary $t$ and $c \in [m] \setminus G_{t-1}$. To simplify the heavy notation, we will omit the subscripts $t-1$ and $t$. For an arbitrary subset $U \subseteq [m]$, define

$$f(U) := \min_{i \in U \setminus \{c\}} \left[ \mathrm{loss}(i, G \cup \{c\}) \right]_{-\infty}^{\Delta}$$

and define $f'$ similarly for a neighboring dataset. Note

$$f(R) = \min \left\{ f(R \cap R'), f(R \setminus R') \right\}$$

and similarly $f'(R') = \min \left\{ f'(R \cap R'), f'(R' \setminus R) \right\}$. The key observation is that, if $i \in R \setminus R'$, then

$$\mathrm{loss}(i, G \cup \{c\}) \in [\Delta - 1, \Delta).$$

So when $R \setminus R'$ is nonempty, we have $f(R \setminus R') \in [\Delta - 1, \Delta)$. Similarly, $f'(R' \setminus R) \in [\Delta - 1, \Delta)$ when $R' \setminus R$ is nonempty.

There are 8 cases depending on whether $R \cap R'$, $R \setminus R'$, and $R' \setminus R$ are empty or not.

Case 1: All of $R \cap R'$, $R \setminus R'$, $R' \setminus R$ are nonempty. The idea is to apply Lemma A.5 repeatedly. Since $\left| \mathrm{loss}(i, G \cup \{c\}) - \mathrm{loss}'(i, G \cup \{c\}) \right| \leq 1$, we have

$$|f(R \cap R') - f'(R \cap R')| \leq 1.$$

Since $f(R \setminus R'), f'(R' \setminus R) \in [\Delta - 1, \Delta]$, we have

$$|f(R \setminus R') - f'(R' \setminus R)| \leq 1.$$

Therefore, $|f(R) - f'(R')| \leq 1$.

Case 2: $R \cap R' \neq \emptyset$, $R \setminus R' = \emptyset$, $R' \setminus R = \emptyset$. That is $R = R'$. This case is subsumed by the first case.

Case 3: $R \cap R' \neq \emptyset$, $R \setminus R' \neq \emptyset$, $R' \setminus R = \emptyset$. That is, $R'$ is a nonempty proper subset of $R$. There are two subcases. First, $f(R \cap R') \leq f(R \setminus R')$, then $f(R) = f(R \cap R') = f(R')$. Therefore,

$$|f(R) - f'(R')| = |f(R') - f'(R')| \leq 1.$$

Second, $f(R \cap R') > f(R \setminus R')$, so $f(R) = f(R \setminus R')$. As $f'(R') \geq f(R') - 1 = f(R \cap R') - 1 > f(R \setminus R') - 1$ and $f'(R') \leq \Delta \leq f(R \setminus R') + 1$. Therefore,

$$|f(R) - f'(R')| = |f(R \setminus R') - f'(R')| \leq 1.$$

Case 4: $R \cap R' \neq \emptyset$, $R \setminus R' = \emptyset$, $R' \setminus R \neq \emptyset$. This case is the same as the third case.

Case 5: $R \cap R' = \emptyset$, $R \setminus R' \neq \emptyset$, $R' \setminus R \neq \emptyset$. That is $R$ and $R'$ are disjoint nonempty sets. In this case, $f(R) = f(R \setminus R') \in [\Delta - 1, \Delta)$ and $f'(R') = f'(R' \setminus R) \in [\Delta - 1, \Delta)$. Thus, $|f(R) - f'(R')| \leq 1$.

Case 6: $R \cap R' = \emptyset$, $R \setminus R' \neq \emptyset$, $R' \setminus R = \emptyset$. That is, $R \neq \emptyset$ and $R' = \emptyset$. As $f'(R') = \Delta$ and $f(R) = f(R \setminus R') \in [\Delta - 1, \Delta)$. Thus, $|f(R) - f'(R')| \leq 1$.

Case 7: $R \cap R' = \emptyset$, $R \setminus R' = \emptyset$, $R' \setminus R \neq \emptyset$. This case is the same as the sixth case.

Case 8: Both $R$ and $R'$ are empty. In this case, $f(R) = f'(R') = \Delta$. $\square$

As the subsampling probability $p_t$ is oblivious to the dataset, replacing $R_t$ by $S_t$ preserves the sensitivity.

**Lemma B.2.** *For every $t$ and every $c \in [m] \setminus G_{t-1}$,*

$$|\mathrm{score}_t(c) - \mathrm{score}'_t(c)| \leq 1.$$

*Proof.* From the proof of Lemma B.1, it suffices to show $S_t \cap (R_t \cap R'_t) = S'_t \cap (R_t \cap R'_t)$. This is true, as we subsample $R_t$ (and $R'_t$) with a fixed probability $p_t$. $\square$

**Lemma B.3** (Privacy). *Algorithm 4 is $\varepsilon$-DP.*

*Proof.* To prove privacy, we analyze the two main stages of the algorithm separately, prove that each stage is $\varepsilon/2$-DP, and then use composition to argue that the final algorithm is $\varepsilon$-DP. For the first stage, note that Lemma B.2 implies that the sensitivity of the function in Equation (6) is at most 1. Therefore, the guarantees of the AboveThreshold algorithm (Lemma A.2) imply that $\tilde{c}_t$ is $\varepsilon_t$-DP. Using DP composition (Lemma A.4), the first stage is $(\sum_t \varepsilon_t)$-DP, i.e. $\varepsilon/2$-DP. For the second stage, following a same proof as in Claim 3.4, the sensitivity of the function in Equation (7) is also at most 1. Therefore its output is $\varepsilon/2$-DP by the guarantees of the exponential mechanism (Lemma A.1). Finally, using DP composition (Lemma A.4), Algorithm 4 is $\varepsilon$-DP. □

Having proved our privacy guarantees, we now move to prove our claim about utility.

**Lemma B.4** (Utility). *Suppose there exist an $i^* \in [m]$ and a sufficiently large constant $C$ such that*

$$\min_{i:\, i \neq i^*} M(i, i^*) \geq \frac{CT^2 \log(Tm/\beta)}{\varepsilon} \,.$$

*Then Algorithm 4 outputs $i^*$ with probability at least $1 - \beta$.*

*Proof.* For every iteration $t \in [T]$, if $i^* \in [m] \setminus G_{t-1}$, then $\tilde{c}_t = \perp$ with probability at most $\beta_t$, by Lemma A.2. Thus, by a union bound, the probability that $i^* \notin G_T$ but $\perp \in G_T$ is at most $\sum_{t=1}^{T} \beta_t = \beta/2$. Our following analysis is conditioned on this bad event does not happen.

There are two cases to consider. The first case is when $i^* \in G_T$. In this case, the exponential mechanism outputs $i^*$ with probability at least $1 - \beta/2$ as long as $\Delta > \frac{\log(2m/\beta)}{\varepsilon/2}$. Let us now consider the second case when $i^* \notin G_T$. Note

$$\Delta_T = \Delta - 16 \sum_{t=1}^{T} \frac{\log(2m/\beta_t)}{\varepsilon_t} \,.$$

Plugging in $\varepsilon_1 = \cdots = \varepsilon_T = \frac{\varepsilon}{2T}$ and $\beta_1 = \cdots = \beta_T = \frac{\min\{\beta/2, 1/m^2\}}{T}$, we have

$$\Delta_T \geq \Delta - \frac{32T^2 \log(4mT/\beta)}{\varepsilon} - \frac{32T^2 \log(2m^3 T)}{\varepsilon} \,.$$

It is not difficult to see $\ell_{i^*} \leq -\Delta$ and $\ell_i \geq \Delta_T$ for every $i \neq i^*$. To ensure the exponential mechanism outputs $i^*$ with probability $1 - \beta/2$, it suffices to ask

$$\Delta_T + \Delta \geq \frac{2 \log(2m/\beta)}{\varepsilon/2} \,.$$

□

**Lemma B.5** (Efficiency). *Algorithm 4 queries at most $O((T + 1)\, m^{1 + \frac{1}{T+1}} \log m)$ entries of $M$ in expectation.*

*Proof.* Let $X$ denote the total number of queries made by the algorithm. In round $t$, the algorithm makes at most $|S_t| \cdot m$ queries. When applying the exponential mechanism, the algorithm makes at most $T \cdot m + |R_{T+1}| \cdot m$ queries. So

$$\mathbb{E}\, X \leq T \cdot m + \mathbb{E}\,|R_{T+1}| \cdot m + \sum_{t=1}^{T} \mathbb{E}\,|S_t| \cdot m$$

$$= \left( T + \mathbb{E}\,|R_{T+1}| + \sum_{t=1}^{T} \mathbb{E}\,|R_t| \cdot p_t \right) \cdot m \,.$$

In the following, we will bound $|R_t|$ for each $t$. Note $|R_1| = m$. Now we bound $|R_t|$ for $t \geq 2$. For $t \in [T]$, let $B_t$ denote the (bad) event that $\tilde{c}_t \neq \perp$ but $\text{score}_t(\tilde{c}_t) < \Delta_t$. For any $k \in \mathbb{N}$, when $B_t$ does not happen, $|R_{t+1}| \geq k$ implies the

following event: there exists $c \in [m] \setminus G_{t-1}$ such that the set $\left\{ i \in [m] : \text{loss}(i, G_{t-1} \cup \{c\}) < \Delta_t \right\}$ has size at least $k$ and none of its element is sampled. Therefore,

$$\Pr\left( |R_{t+1}| \geq k \,\middle|\, \overline{B_t} \right) \leq \min\left\{ m(1 - p_t)^k, 1 \right\}.$$

Let $K := \max\left\{ k \in \mathbb{Z}_{\geq 0} : m(1 - p_t)^k \geq 1 \right\}$. Note $K = \left\lfloor \frac{\ln m}{\ln 1/(1-p)} \right\rfloor$. Then

$$\mathbb{E}\left[ |R_{t+1}| \,\middle|\, \overline{B_t} \right] = \sum_{k=1}^{\infty} \Pr\left( |R_{t+1}| \geq k \,\middle|\, \overline{B_t} \right)$$

$$\leq K + 1 + \frac{m(1 - p_t)^{K+1}}{p_t}$$

$$\leq 2 + \frac{1 + \ln m}{p_t} \leq \frac{2 \ln m}{p_t}.$$

Let $B$ be the (bad) event some $B_t$ happens, i.e. $B := \cup_t B_t$. Using the guarantees of AboveThreshold (Lemma A.2) and a union bound, we have

$$\Pr(B) \leq \sum_{t=1}^{T} \Pr(B_t) \leq \sum_{t=1}^{T} \beta_t.$$

As $\beta_t \leq (1/m^2)/T$ for all $t$, we have $\Pr(B) \leq 1/m^2$.

Thus, $\mathbb{E}[|X| \,|\, \bar{B}]$ is upper bounded by

$$2m \ln m \cdot \left( \frac{1}{p_T} + p_1 m + \frac{p_2}{p_1} + \frac{p_3}{p_2} + \cdots + \frac{p_T}{p_{T-1}} \right) + Tm. \tag{8}$$

Plugging in[2] $p_t = m^{-1 + \frac{t}{T+1}}$ for $t \in [T]$, we have

$$\mathbb{E}[|X| \,|\, \bar{B}] \leq 2(T + 1) \cdot m^{1 + \frac{1}{T+1}} \ln m + Tm.$$

Finally,

$$\mathbb{E}\, X = \mathbb{E}[X \,|\, \bar{B}] \Pr(\bar{B}) + \mathbb{E}[X \,|\, B] \Pr(B)$$

$$\leq \mathbb{E}[X \,|\, \bar{B}] + m^2 \cdot \Pr(B)$$

$$\leq 2(T + 1) \cdot m^{1 + \frac{1}{T+1}} \ln m + 1 + Tm.$$

$\square$

## C. Proofs for Section 4.1

We prove Theorem 4.2 in this section. We start with the simpler case where $T = 1$ and the extension to general $T$ is straightforward. Given a dataset $x_{1:n} = \{x_1, \ldots, x_n\}$, we define the matrix $M$ in the same way as Theorem 4.1 up to a scaling factor of $\frac{n}{4\varepsilon}$,

$$M(i, j) = M(i, j; x_{1:n}) := 2 \sum_{k=1}^{n} \frac{1}{n} \left[ \log \frac{H_j(x_k)}{H_i(x_k)} \right]_{-\varepsilon}^{\varepsilon} + \mathbb{E}_{H_i} \left[ \log \frac{H_i(X)}{H_j(X)} \right]_{-\varepsilon}^{\varepsilon} - \mathbb{E}_{H_j} \left[ \log \frac{H_j(X)}{H_i(X)} \right]_{-\varepsilon}^{\varepsilon}.$$

Applying Algorithm 3 directly to this matrix $M$ does not guarantee utility anymore, as we no longer have a hypothesis $H_{i^*}$ that explains the dataset better than any other hypothesis. In particular, the AboveThreshold step does not guarantee to output a good player w.h.p.. The reason is that, in addition to $H_{i^*}$, there could be other hypotheses $H_i$ also close to $P$, so

---

[2]Applying AM-GM inequality to Equation (8) will show these are the optimal values for $\{p_t\}$.

there does not exist an $i$ such that $\max_j M(i, j)$ is small. Fortunately, we are in a win-win situation. If there exists some $i$ in the random subset $S$ with small $\max_j M(i, j)$, then we know $H_i$ is close to $P$ and we can just return it (see G2 of Lemma C.1). Otherwise, if every $i \in S$ has large $\max_j M(i, j)$, then we know $M(i, i^*)$ is also large for very $i \in S$ (see G3 of Lemma C.1). In this case, the AboveThreshold step is guaranteed to output a good player w.h.p. and we can proceed to the exponential mechanism as before. The algorithm is summarized in Algorithm 5.

---

**Algorithm 5** Modified Sample-and-Prune Algorithm

---

**Parameters**: Privacy parameter $\varepsilon$, failure probability $\beta$, TV bound $\eta$

$\tau_A \leftarrow C_A \varepsilon \eta$
$\tau_B \leftarrow C_B \varepsilon \eta$
$\tau_E \leftarrow C_E \varepsilon \eta$

*// Find a Good Player*
$p \leftarrow 1/\sqrt{m}$
$S \leftarrow$ Subsample $[m]$ with probability $p$
**For** $i \in S$, $\text{loss}(i) \leftarrow \max_j M(i, j)$
$c_B \leftarrow \text{BelowThreshold}\big(\{\text{loss}(i)\}_{i \in S}, \varepsilon/3, \tau_B\big)$
**If** $c_B \neq \perp$ **then return** $c_B$
**For** $c \in [m]$, $\text{score}(c) \leftarrow \min_{i \in S \setminus \{c\}} M(i, c)$
$c_A \leftarrow \text{AboveThreshold}\big(\{\text{score}(c)\}_{c \in [m]}, \varepsilon/3, \tau_A\big)$
**If** $c_A = \perp$ **then abort**

*// Exponential Mechanism*
**For** $i \in [m]$,

$$
\ell_i \leftarrow \begin{cases} \left[\displaystyle\max_{j:j \neq i} M(i, j)\right]_{-\infty}^{\tau_E}, & M(i, c_A) < \tau_E \\ M(i, c_A), & M(i, c_A) \geq \tau_E \end{cases} \tag{9}
$$

Sample a player $i \in [m]$ with probability $\propto e^{-\frac{\varepsilon}{6} \cdot \ell_i}$

---

**Privacy**  Since BelowThreshold, AboveThreshold, and the exponential mechanism are $\varepsilon/3$-DP respectively, the privacy guarantee follows directly from the composition lemma.

**Utility**  The utility analysis replies on the following lemma which is implicit in (Asi et al., 2024).

**Lemma C.1.** *Let* $\mathcal{H} := \{H_1, \ldots, H_m\}$ *be a set of distributions. Let* $x_1, \ldots, x_n$ *be drawn from a distribution* $P$ *such that* $\|P - H_{i^*}\|_{\text{TV}} \leq \eta$ *for some* $i^* \in [m]$. *For* $\varepsilon \leq O(\eta)$ *and* $n \gtrsim \frac{\log(m/\beta)}{\varepsilon \eta}$, *the following good events hold with probability at least* $1 - \beta$ *for sufficiently large constants* $C_1, c_2, C_2, C_3, c_3$ *satisfying* $c_2 < C_2$ *and* $c_3 < C_3$.

*G1:* $\max_j M(i^*, j) \leq C_1 \varepsilon \eta$;

*G2: If* $\max_j M(i, j) \geq C_2 \varepsilon \eta$, *then* $M(i, i^*) \geq c_2 \varepsilon \eta$.

*G3: If* $\max_j M(i, j) \leq c_3 \varepsilon \eta$, *then* $D_\varepsilon(H_i, P) \leq C_3 \varepsilon \eta$;

*Proof.*

**G1**  Recall the notation $\text{loss}(i) = \max_j M(i, j)$. By Lemma 15 of (Asi et al., 2024), for any $B \geq 4\varepsilon\eta$, we have $\text{loss}(i^*) \leq B$ with probability at least $1 - m \exp(-\Omega(nB))$.

**G2**  Let us prove the contrapositive. Assume $M(i, i^*) \leq c_2 \varepsilon \eta$ and our goal is to show $\max_j M(i, j) \leq C_2 \varepsilon \eta$ for some constant $C_2 = C_2(c_2) \geq c_2$. There are two cases.
Case 1 : $D_\varepsilon(H_i, H_{i^*}) \geq 8\varepsilon\eta$. By Lemma 41 of (Asi et al., 2024), $D_\varepsilon(H_i, H_{i^*}) - M(i, i^*) \leq D_\varepsilon(H_i, H_{i^*})/2$ with

probability at least $1 - \exp\left(-\Omega(n\varepsilon\eta)\right)$. Then, $D_\varepsilon(H_i, H_{i^*}) \leq 2c_2\varepsilon\eta$.

Case 2: $D_\varepsilon(H_i, H_{i^*}) < 8\varepsilon\eta$. By Lemma 24 of (Asi et al., 2024),

$$D_\varepsilon(H_i^*, P) \leq 2\varepsilon\|H_i^* - P\|_{\text{TV}} \leq 2\varepsilon\eta\,.$$

By Lemma 27 of (Asi et al., 2024),

$$D_\varepsilon(H_i, P) \lesssim D_\varepsilon(H_i, H_i^*) + D_\varepsilon(H_i^*, P) \lesssim \varepsilon\eta\,.$$

By Lemma 25 of (Asi et al., 2024),

$$\|H_i - P\|_{\text{TV}} \leq \frac{D_\varepsilon(H_i^*, P)}{2\varepsilon} + \frac{e^\varepsilon - 1}{2}\,.$$

Thus, given $\varepsilon \leq O(\eta)$, we have $\|H_i - P\|_{\text{TV}} \leq O(\eta)$. By Lemma 15 of (Asi et al., 2024), $\max_j M(i, j) \leq C_2\varepsilon\eta$ for sufficiently large constant $C_2$.

**G3**  Assume $\max_j M(i, j) \leq c_3\varepsilon\eta$ for a large enough constant $c_3$. Then $M(i, i^*) \leq c_3\varepsilon\eta$. By Lemma 41 of (Asi et al., 2024), $D_\varepsilon(H_i, H_{i^*}) \leq 2c_3\varepsilon\eta$. By Lemma 27 of (Asi et al., 2024),

$$D_\varepsilon(H_i, P) \lesssim D_\varepsilon(H_i, H_{i^*}) + D_\varepsilon(H_{i^*}, P) \lesssim \varepsilon\eta\,.$$

$\square$

Now we prove the utility of Algorithm 5. We condition our analysis on that (1) the good events in Lemma C.1 happen; (2) BelowThreshold succeeds; (3) AboveThreshold succeeds.

If $c_B \neq\perp$, then

$$\max_j M(c_B, j) \leq \tau_B + O\left(\tfrac{\log(m/\beta)}{n}\right) \leq (C_B + 1)\varepsilon\eta\,.$$

where the first inequality follows from Lemma A.3, and the second inequality follows by setting $\tau_B = C_B\varepsilon\eta$ and using $n \gtrsim \frac{\log(m/\beta)}{\varepsilon\eta}$. Then by Lemma C.1, $D_\varepsilon(H_{c_B}, P) \leq O(\varepsilon\eta)$.

Otherwise $c_B =\perp$, then for any $i \in S$ we have

$$\max_j M(i, j) \geq \tau_B - O\left(\tfrac{\log(m/\beta)}{n}\right) \geq (C_B - 1)\varepsilon\eta\,.$$

Then by Lemma C.1, we have for any $i \in S \setminus \{i^*\}$,

$$M(i, i^*) \geq c\varepsilon\eta \geq \tau_A + O\left(\tfrac{\log(m/\beta)}{n}\right)$$

by setting $\tau_A = C_A\varepsilon\eta$ for some proper constant $C_A$. Then by Lemma A.2, $c_A \neq\perp$ and for any $i \in S \setminus \{c_A\}$,

$$M(i, c_A) \geq \tau_A - O\left(\tfrac{\log(m/\beta)}{n}\right) \geq (C_A - 1)\varepsilon\eta\,.$$

By Lemma C.1, $\max_j M(i^*, j) \leq C_1\varepsilon\eta$. Choose $C_A, C_1$ such that $C_A - 1 \geq C_1 + 1$. So the exponential mechanism will return an $i$ such that

$$\max_j M(i, j) \leq \max_j M(i^*, j) + O\left(\tfrac{\log(m/\beta)}{n}\right) \leq (C_1 + 1)\varepsilon\eta\,.$$

Then by Lemma C.1, $D_\varepsilon(H_i, P) \leq O(\varepsilon\eta)$.

**Efficiency**  Compared with Algorithm 3, note that the additional number of queries is incurred by BelowThreshold, which is the same as AboveThreshold. Therefore, the query complexity increases by at most a factor of 2.

