# OpenReview forum: "Fast and Near-Optimal Algorithms for Private Hypothesis Selection"
_ICML.cc/2026/Conference — ICML 2026 regular_

### Official Review · Reviewer_bwDo · 2026-03-09

**Soundness:** 4
**Presentation:** 4
**Significance:** 4
**Originality:** 4
**Overall Recommendation:** 5
**Confidence:** 4

**Summary:**

This work considers the problem of private hypothesis selection. Given $m$ hypotheses $H_1, \dots, H_m$ and sample access to an unknown distribution, find the hypothesis that best fits the observed samples. Formally, (in the "simplified" setting), we are given sample access to $H_{i*}$ and asked to return $i*$. In the private hypothesis selection problem, our algorithm must be differentially private: for any two neighboring datasets (i.e. differing at one point), the probability of any set of outputs differs by a multiplicative factor of $\exp(\varepsilon)$. Previous algorithms for the problem either incurred sub-optimal sample complexity e.g. $O(1/\varepsilon^2)$ or sub-optimal run-time e.g. $O(m^2 n)$. This works shows that it is possible to simultaneously achieve the best of both worlds: providing $O(1/\varepsilon)$ sample complexity and $O(mn)$ run-time.

A standard (sample-optimal) algorithm is as follows: compare every pair of hypotheses on the dataset in $O(m^2n)$ time, and sample a winner from this tournament using the exponential mechanism. The key step of the paper's algorithm is to first prune the set of players to establish a set of good players, and then only compare players against this smaller set. For example, if we sample a set of $\sqrt{m}$ players and find a player that beats all of them (guaranteed to exist since the tournament has a winner), then at most $\sqrt{m}$ players can beat this player. In particular, we can essentially perform the exponential mechanism on $\sqrt{m}$ players that beat the chosen good player, obtaining $O(m^{1.5} n)$ run-time. The authors then improve upon this basic scheme by adding more "levels" to the algorithm, obtaining $O(m^{1 + 1/T} n)$ for increasingly large values of $T$. A crucial observation in this scheme to preserve privacy is that among the set of players that survive pruning, their scores do not change significantly: in particular, any player that survives in one dataset but not the other must be "nearly" defeated by the surviving player.

**Compliance With Llm Reviewing Policy:**

Affirmed.

**Key Questions For Authors:**

Is there any fundamental obstacle to achieving $O(m + n)$ run-time? What is the best sample complexity one can achieve under this constraint?

Is hypothesis selection the best algorithm for solving universal statistical estimation? For example, if the map $\Theta$ is simple (eg mean estimation), there is no reason to first construct a cover.

Is there an intuitive reason for why capping by $\varepsilon$ is useful? Is it because after that threshold, it does not matter how much one hypothesis better explains the data?

**Limitations:**

yes

**Strengths And Weaknesses:**

**Strengths**

Private hypotheses testing is a natural problem to study, and the authors obtain optimal sample complexity and state-of-the-art run-time, and is the first algorithm to simultaneously achieve both.

The algorithm is clear and well-written, and the ideas are presented nicely.

**Weaknesses**

No particularly obvious weaknesses. One point is that the algorithm for private universal statistical estimation still depends on the size of a cover, which can be unreasonably large in some cases, but this is not the main result of the paper.

---

> ### Author Rebuttal · Authors · 2026-03-31
>
> We thank the reviewer for constructive feedback and overall positive evaluation!
>
> Regarding the question about achieving $O(m + n)$ run-time, the best possible running time one can hope for is $\Theta(m n)$, as we need to check each of the $m$ hypotheses at least once and each hypothesis takes at least $n$ time.
>
> Regarding the question about universal statistical estimation, we agree that hypothesis selection is not always the most efficient approach for every estimation problem. In particular, for simple tasks such as mean estimation, there are direct algorithms that avoid constructing a cover and achieve optimal guarantees more efficiently.
> Our focus is on the general problem of private estimation over arbitrary model classes, where hypothesis selection provides a generic and broadly applicable framework. In such settings, especially when the structure of the problem does not admit specialized estimators, reductions to hypothesis selection are standard and often unavoidable. Our contribution is to show that this general approach can be made computationally efficient while retaining near-optimal statistical guarantees.
> We will clarify this distinction in our revision.
>
> Regarding "capping by $\epsilon$", the main motivation is to limit the influence of a single datapoint on the log-likelihood ratio. Without such a capping, it is impossible to make the log-likelihood ratio private, as its sensitivity would be unbounded.

---

> > ### Author Rebuttal · Reviewer_bwDo · 2026-04-02
> >
> > I thank the author for their answers and remain positive in my assessment.
> >
> > Regarding  a possible runtime of $O(m + n)$, is it not possible (theoretically) that one could have an algorithm that reads the dataset once and reads each hypothesis once? Or is the $O(mn)$ already due from reading each hypothesis once (say if the hypothesis itself is defined not as some simple function, but as how it evaluates on each point in the dataset)? In the latter case, a $O(m + n)$ algorithm seems possible (although certainly not at all clear how to achieve), except perhaps in some special cases.

---

> > > ### Author Response · Authors · 2026-04-04
> > >
> > > We thank the reviewer for this insightful question. While one might hope for an O(m+n)-time algorithm that reads the data and each hypothesis only once, this is not possible for general hypothesis classes.
> > >
> > > The key issue is that, in the absence of additional structure, selecting a good hypothesis requires estimating its loss (or score) on the dataset. This inherently involves evaluating each hypothesis on many (in the worst case, all) data points. From an information-theoretic perspective, if two hypotheses differ only on a small subset of the data, any algorithm must examine those data points to distinguish between them; thus, one cannot avoid a dependence on mn in the worst case.

---

### Official Review · Reviewer_q3dp · 2026-03-11

**Soundness:** 3
**Presentation:** 3
**Significance:** 3
**Originality:** 3
**Overall Recommendation:** 4
**Confidence:** 3

**Summary:**

This paper considers improving the time complexity while maintaining the near-optimal sample complexity for the private hypothesis selection problem. Extensions and implications for the agnostic setting, where the true hypothesis is not in the candidate set, and for universal estimation are also studied.

**Compliance With Llm Reviewing Policy:**

Affirmed.

**Final Justification:**

I am still in favor of accepting this work, but I have adjusted my rating to reflect the lack of experimental analysis.

**Key Questions For Authors:**

The only suggestion that I have is perhaps given the main selling point being a fast algorithm, is it possile to add some simulations to verify the computational gains while maintaing the near-optimal sample complexity.

**Limitations:**

yes

**Strengths And Weaknesses:**

This is a technically strong paper with well-motivated methodology and solid theoretical results. I don't see any imminent weakness.

---

> ### Author Rebuttal · Authors · 2026-03-31
>
> We thank the reviewer for constructive feedback and overall positive evaluation!
>
> We thank the reviewer for the suggestion regarding synthetic experiments. Our goal in this work is to establish computational and statistical complexity tradeoffs of private hypothesis selection, and in particular to show that near-linear-time algorithms are possible while maintaining near optimal guarantees. As such, we focused on the theoretical aspects and did not include an experimental evaluation. That said, we agree that empirical results could help illustrate the practical gains, and we will consider including a small synthetic experiment in a future version.

---

> > ### Author Rebuttal · Reviewer_q3dp · 2026-04-01
> >
> > Thanks.

---

### Official Review · Reviewer_jB3G · 2026-03-11

**Soundness:** 3
**Presentation:** 3
**Significance:** 2
**Originality:** 2
**Overall Recommendation:** 4
**Confidence:** 3

**Summary:**

This paper studies $\epsilon$-differentially private hypothesis selection over a finite class $\mathcal H=\{H_1,\dots,H_m\}$. In the simple setting, the data are drawn from some \(H_{i^\star}\in\mathcal H\), and the goal is to identify \(i^\star\) privately with probability at least $1-\beta$. The paper also considers an agnostic setting in which the true distribution is only close to one of the hypotheses.

The main technical ingredient is a new algorithm for the private tournament problem that replaces the standard exhaustive pairwise comparison approach with a two-stage procedure. First, the algorithm privately identifies a small set of “strong” hypotheses using subsampling and the AboveThreshold algorithm. Then, it runs the exponential mechanism with a carefully designed truncated loss that can be evaluated much more cheaply for most candidates. This leads to a $T$-round algorithm for simple hypothesis selection with sample complexity $O(T^2 \log(Tm/\beta)/\Delta_\varepsilon(\mathcal H))$ and expected running time $O(T n m^{1+\frac{1}{T+1}}\log m)$. The paper also gives an extension to the agnostic setting in the high-privacy regime, a lower bound showing that the \(\log m\) dependence is unavoidable on some instances, and an application to faster private universal statistical estimation.

**Compliance With Llm Reviewing Policy:**

Affirmed.

**Final Justification:**

The rebuttal addressed my questions. The lack of experimental analysis is the only reason I maintain a *weak* accept.

**Key Questions For Authors:**

- Is there a principled recommendation for choosing $T$ in practice?

- In Theorem 4.2, how essential is the assumption $\varepsilon \le O(\eta)$? Can the approach be extended beyond the high-privacy regime?

- Is the lower bound also true beyond disjoint support distributions?

 - I am a bit unsure about one step in the privacy proof of Algorithm 3. In Lemma B.2, the argument seems to rely on the fact that
  $
  S_t \cap (R_t \cap R_t') = S_t' \cap (R_t \cap R_t'),
  $
  where $S_t$ and $S_t'$ are obtained by subsampling $R_t$ and $R_t'$ with probability $p_t$. Could the authors clarify this step in more detail? As currently written, it is not fully clear to me why subsampling with the same inclusion probability is enough to guarantee this identity.

**Limitations:**

There is no limitations section in the paper.

**Strengths And Weaknesses:**

Strengths:
- The paper revisits an important problem and addresses a real gap in the literature
- The paper is well written, and the intuitions and proofs are well explained
- The main algorithmic idea is intuitive, yet not trivial, and the results achieve a compelling tradeoff between computational and statistical efficiency

Weaknesses:
- The main weakness is the complete lack of empirical evaluation. Since the contribution is largely about improving computational efficiency while keeping essentially the same statistical complexity, I think even a small synthetic experimental section would have helped a lot. In particular, it would be useful to see whether the multi-round procedure actually gives meaningful acceleration over Algorithm 1 / the baseline tournament-style approaches for realistic values of $m$, and whether the hidden constants in the $O$ theoretical bounds are benign.
- The role of the parameter $T$ is not fully clarified from a practical perspective. The theory gives a tradeoff, but the paper does not really discuss how one should choose $T$ as a function of $m$, $n$, or computational budget.
- The agnostic result is limited to the high-privacy regime, and the lower bound is only true for distributions with disjoint support

---

> ### Author Rebuttal · Authors · 2026-03-31
>
> We thank the reviewer for their constructive feedback and overall positive evaluation!
>
> We thank the reviewer for the suggestion regarding synthetic experiments. Our goal in this work is to establish computational and statistical complexity tradeoffs of private hypothesis selection, and in particular to show that near-linear-time algorithms are possible while maintaining near optimal guarantees. As such, we focused on the theoretical aspects and did not include an experimental evaluation. That said, we agree that empirical results could help illustrate the practical gains, and we will consider including a small synthetic experiment in a future version.
>
> Regarding the assumption $\epsilon \le O(\eta)$ in Theorem 4.2 (i.e., the application of our private tournament algorithm to the case where the true distribution is guaranteed to be eta close to the hypothesis set in TV distance), it does not seem to be removable with our current proof technique, as this assumption plays an important role in our utility analysis. We expect a rather different proof technique or algorithmic idea would be needed to get rid of this assumption.
>
> Regarding lower bounds, we acknowledge that our lower bound is only a worst-case lower bound and we will clarify that in the intro. An instance-optimal lower bound would be the ultimate goal, but such a lower bound is not even known in the non-private case in the multiple hypothesis case with $m>2$. So in order to get an instance-optimal private lower bound, one would first solve the non-private problem, which is an interesting open question.
>
> Regarding the privacy proof of Algorithm 3, $R_t$ can be decomposed into $R_t \cap R'_t$ and $R_t \setminus R'_t$. Similarly, $R’_t$ can be decomposed into $R’_t \cap R_t$ and $R’_t \setminus R_t$. Following the proof of Lemma 8.1, we only need to guarantee that everything subsampled from $R_t \cap R'_t$ is the same as what is subsampled from $R’_t \cap R_t$. This is straightforward, as these two sets (i.e., $R_t \cap R'_t$ and $R’_t \cap R_t$) are identical, and the subsampling probabilities are the same.

---

> > ### Author Rebuttal · Reviewer_jB3G · 2026-03-31
> >
> > I thank the authors for their rebuttal and maintain my positive evaluation of the paper.

---

### Official Review · Reviewer_7inp · 2026-03-20

**Soundness:** 3
**Presentation:** 3
**Significance:** 3
**Originality:** 3
**Overall Recommendation:** 4
**Confidence:** 4

**Summary:**

This paper is about private hypothesis selection, which is a very well studied model by now in differential privacy. The setting is as follows: we have a family of $m$ distributions $\mathcal{H} = \set{ H_1, \dots, H_m }$ and we have a set of $n$ samples and want to find out which one of the $H_i$'s produced them (or are the best fit for them) using as few samples as possible. They consider the realizable setting and the agnostic setting separately: the distribution that generates the $n$ samples could be from $\mathcal{H}$ or be within tv-distance $\eta$ from one of them. This problem has been studied before. However, no prior work has been able to obtain $O(\log m)$ sample complexity together with a linear in $m$ runtime, which is the focus of this work. In the realizable setting they show that for any $T$ there exists an algorithm that takes in $O(T^2 \log(Tm)) / \Delta_\epsilon(\mathcal{H})$ many samples and the correct distribution and has runtime in expectation $\tilde{O}(Tnm^{1 + \frac{1}{T+1}} )$. So by appropriately choosing $T$ (such as $\log \log m$) one may approach the $\log m$ sample complexity and $mn$ runtime. The result for the agnostic case is also similar. Note that the $\Delta_\epsilon(\mathcal{H})$ is a notion of divergence present in prior work of (Asi et al. 2024). They also show that the sample complexity bound is tight whenever $\mathcal{H}$ has disjoint distributions.

Technically, they reduce the problem to a private "tournament winner" selection problem from a matrix, that does not leak too much privacy, and has fast (here it means smaller than $m^2$) runtime. Similar ideas have been considered before (Aden-Ali et al., 2021;Asi et al., 2024). The main contribution of this work is an algorithmic idea to use an iterative approach, ran for $T$ steps, where in each step, a set of the "players" are subsampled, a winner amongst them is chosen, and then there is an aggregation step that using an exponential mechanism with the right score function samples the best player from the winners of different rounds.

**Compliance With Llm Reviewing Policy:**

Affirmed.

**Final Justification:**

I am maintaining my score. The paper gives a technically solid improvement for private hypothesis selection. My main concern was instance optimality of the lower-bound, and the rebuttal addressed this by clarifying that such lower bounds are not even known even non-privately. Overall I think this is a nice paper with a novel algorithmic idea and sufficient technical merit for acceptance.

**Key Questions For Authors:**

Comments: Not too important but $\lVert \cdot \rVert_{\infty}$ used in the beginning of Section 3 usually refers to the maximum row norm of a matrix, whereas here it is used to refer to the maximum entry wise distance. The standard notation for that is $\lVert \cdot \rVert_{\max}$.

**Limitations:**

yes

**Strengths And Weaknesses:**

Soundness: I have not noticed any major issues with the correctness of the upper bounds and algorithms. That being said, the presentation of the lower bound in the intro can be misleading, if the claim is that $\log(m/\beta) / D_\epsilon(H)$ many samples are necessary, that should be true for any choice of $H$ (in a sense, instance-optimality) not just disjoint $H$ where it holds from the $\log(m / \beta) / \epsilon$ lower bound. That leaves us with the $O(1 / D_\epsilon(H))$ bound from (Asi et al., 2023).

Presentation: The paper is well written.

Significance: The problem of hypothesis selection is perhaps one of the most fundamental problems in differential privacy. Because of the lower bound issue highlighted above, I don't think the problem is yet "fully" solved in an instance optimal sense, but I think this is a nice improvement on it. In terms of practical impact, It's not clear to me currently if this algorithm is practical or not, which hopefully can be clarified by the authors.

Originality: The main novelty of this work is the subsampling and "selecting the best in subsample" algorithmic idea to avoid checking every hypothesis and improve the runtime. I expect similar ideas might have been appeared in other areas (maybe not in privacy) to improve runtimes. I think the application of it to this setting is novel and putting the pieces together in a way that works with privacy takes some effort (e.g. setting up the right exponential mechanism).

Overall I lean towards accepting this paper.

---

> ### Author Rebuttal · Authors · 2026-03-31
>
> We thank the reviewer for their constructive feedback and overall positive evaluation!
>
> Regarding lower bounds, we acknowledge that our lower bound is only a worst-case lower bound and we will clarify that in the intro. An instance-optimal lower bound would be the ultimate goal, but such a lower bound is not even known in the non-private case in the multiple hypothesis case with $m>2$. So in order to get an instance-optimal private lower bound, one would first solve the non-private problem, which is an interesting open question.
>
> We thank the reviewer for their question regarding practicality. Our work is primarily theoretical, aiming to understand the computational complexity of private hypothesis selection. Prior optimal approaches (e.g., exponential mechanism or tournament-based methods) typically require quadratic or near-quadratic time in the number of hypotheses $m$, which is prohibitive at scale. In contrast, our algorithm runs in near-linear time $m^{1+o(1)}$ while retaining (near-)optimal statistical guarantees, showing that optimal private selection can in fact be computationally efficient.
>
> Regarding $\| \cdot \|_\infty$, thanks for pointing this out, we will change to the more standard notation $\|\cdot\|_\max$.

---

> > ### Author Rebuttal · Reviewer_7inp · 2026-04-04
> >
> > Thank you for the helpful rebuttal and clarifications. I appreciate the explanation on the lower bounds. On practicality, I understand the paper is primarily theoretical, my only point was that, if the authors also see practical relevance, that could be worth highlighting. Overall, I still think this is a nice paper and I am maintaining my score.

---

### Decision · Program_Chairs · 2026-04-30

**Decision:**

Accept (regular)

**Comment:**

Reviewers all agreed that the paper considered a natural problem and clearly explained an interesting solution that meaningfully improved the current state of the art. The authors should make sure to incorporate the (largely minor) clarifying suggestions offered by the reviewers in the final version.